# The Role of Neuropeptide Y in Adipocyte-Macrophage Crosstalk during High Fat Diet-Induced Adipose Inflammation and Liver Steatosis

**DOI:** 10.3390/biomedicines9111739

**Published:** 2021-11-22

**Authors:** Seongjoon Park, Toshimitsu Komatsu, Hiroko Hayashi, Ryoichi Mori, Isao Shimokawa

**Affiliations:** Department of Pathology, Nagasaki University School of Medicine, Graduate School of Biomedical Sciences, 1-12-4 Sakamoto, Nagasaki 852-8523, Japan; komatsut@nagasaki-u.ac.jp (T.K.); hayashih@nagasaki-u.ac.jp (H.H.); ryoichi@nagasaki-u.ac.jp (R.M.); shimo@nagasaki-u.ac.jp (I.S.)

**Keywords:** obesity, fatty liver, neuropeptide Y, macrophage, brown adipose tissue

## Abstract

Obesity is associated with an increased risk of non-alcoholic fatty liver disease (NAFLD), which is initiated by adipocyte-macrophage crosstalk. Among the possible molecules regulating this crosstalk, we focused on neuropeptide Y (NPY), which is known to be involved in hypothalamic appetite and adipose tissue inflammation and metabolism. In this study, the NPY^−/−^ mice showed a marked decrease in body weight and adiposity, and lower free fatty acid and adipose inflammation without food intake alteration during a high fat diet (HFD). Moreover, NPY deficiency increased the thermogenic genes expression in brown adipose tissue. Notably, NPY-mRNA expression was upregulated in macrophages from the HFD mice compared to that from the mice on a standard diet. The NPY-mRNA expression also positively correlated with the liver mass/body weight ratio. NPY deletion alleviated HFD-induced adipose inflammation and liver steatosis. Hence, our findings point toward a novel intracellular mechanism of NPY in the regulation of adipocyte-macrophage crosstalk and highlight NPY antagonism as a promising target for therapeutic approaches against obesity and NAFLD.

## 1. Introduction

Fat is an indispensable component of the cell membrane and acts as an insulator and energy source in mammals [1]. However, excessive fat accumulation may lead to obesity and impaired metabolic homeostasis [2]. Obesity is a growing epidemic and is associated with several metabolic disorders, such as hypertension, hyperglycemia, and dyslipidemia, collectively termed metabolic syndrome [3,4]. The prevalence of obesity in adult as well as children has increased dramatically over the last several decades [5]. The non-alcoholic fatty liver disease (NAFLD), characterized by a progressive fat deposition in the liver, is a consequence of obesity in all ages and may increase the incidence of more serious liver dysfunction, such as non-alcoholic steatohepatitis, fibrosis, and cirrhosis [6]. Hepatic lipid derived from a fat diet and de novo lipogenesis within hepatocytes. However, hepatic lipid is mainly results from adipose tissue dysfunction caused by insulin resistance-mediated lipolysis in patient with NAFLD [7]. Lipid accumulation in the liver, caused by an imbalance between lipid availability and disposal, can lead to oxidative stress, metabolic inflammation, and hepatocyte injury [8].

The crosstalk between adipocytes and macrophages within white adipose tissue (WAT) influences obesity and related metabolic disorders. Macrophage infiltration in WAT during obesity is closely linked to increased free fatty acid (FFA) concentration because macrophages are surrounded by adipocytes that constantly release FFA via lipolysis. Thus, FFA can potentially activate adipose tissue macrophages and consequently alter their function. FFA can activate of M1 (pro-inflammatory) macrophages, which leads to the secretion of inflammatory mediators, including TNFα, IL6, and IL-1β in the adipose tissue, liver, and skeletal muscle, thereby inducing cellular inflammation and insulin resistance [9].

NPY is a crucial factor that regulates appetite and energy expenditure in the brain [10]. NPY increases food intake and promotes adiposity when administered exogenously or overexpressed [11,12]. Conversely, suppression of NPY decreases weigh gain and adiposity in obese mice [13,14]. We recently found that NPY regulates fat availability and disposal by controlling lipogenesis and lipolysis [15,16]. This supports the hypothesis that NPY might be critical in controlling obesity and related diseases. In this study, we investigate the effects of NPY inhibition on of adipocytes-macrophage crosstalk regulation in high fat diet (HFD)-induced obese mice. We found that NPY-mRNA expression in macrophages positively correlated with liver weight/body weight (BW) ratio. Furthermore, deletion of NPY reduced adiposity and body weight gain caused by HFD, enhanced brown adipose tissue (BAT) thermogenesis, inhibited TNFα-mediated adipose tissue inflammation and controlled CD36-mediated fatty acid uptake. These findings suggest that macrophage-released NPY may be a novel therapeutic target for treating obesity and NAFLD.

## 2. Materials and Methods

### 2.1. Animals

The animal care and experimental protocols were approved by the Ethics Review Committee for Animal Experimentation at Nagasaki University. Male NPY^−/−^ mice (129S-Npytm1Rpa/J) and female NPY^+/+^ mice (129S6/SvEvTac) were obtained from Jackson Laboratory (Bar Harbor, ME, USA) and Taconic Farms, Inc. (Germantown, NY, USA). They were bred in a barrier facility at the Center for Frontier Life Sciences at Nagasaki University. NPY-null allele maintained on a mixed genetic background derived from intercrosses between the NPY^+/+^ mice (129S6/SvEvTac) and NPY^−/−^ mice (129S-Npytm1Rpa/J). Three mice were housed in individual cages in the barrier facility (temperature, 21–24 °C; 12 h light/dark cycle) under specific pathogen-free conditions, which were maintained for the entire study. All mice were fed ad libitum (AL) with Charles River Formula 1 (CRF-1) diet (Oriental Yeast Co. Ltd., Tokyo, Japan). At 12 weeks of age, mice were divided into CRF-1 and HFD (60% kcal fat) groups. Body weight was monitored every 4 weeks. At the 6-month old, percentage of body fat was measured using 3D micro-CT (Rigaku Co., Tokyo, Japan). After serum samples collected from cardiac puncture, mice were sacrificed, and tissues were immediately collected.

### 2.2. Metabolic Studies

Whole-body O_2_ consumption and CO_2_ production were monitored using an O_2_/CO_2_ metabolism measuring system for small animals (MK-5000RQ, Muromachi Kikai Co, Tokyo, Japan). In Glucose tolerance test (GTT) experiments, male mice were injected intraperitoneally with glucose (1 g per kg body weight) after a 4 h fast. Blood was drawn from the tail vein at 0, 15, 30, 60 and 120 min and glucose levels were measured using an Accu-check Aviva Nano blood glucose meter (Roche Diagnostics, Tokyo, Japan).

### 2.3. Insulin and FFA Quantification

Serum insulin concentrations were determined using ELISA system (Morinaga institute of biological science, Inc., Yokohama, Japan). Serum FFA level was measured with commercially available assay kit (WAKO pure chemical industries, Ltd., Osaka, Japan).

### 2.4. Liver Triglyceride (TG) Content

Liver samples (50 mg) were homogenized in 1 mL 5% NP-40 solution using a tissue homogenizer. Homogenates were heated at 80–100 °C in a water bath for 5 min and then cool down to room temperature. Homogenates were heat one more time to solubilize all triglyceride and then centrifuge for 2 min at 10,000× *g*. The supernatants were collected, and TG content determined using the Triglyceride Determination Kit (Wako Pure Chemical Industries, Osaka, Japan).

### 2.5. Quantitative Real-Time PCR

Total RNA was isolated using RNeasy tissue kit (Qiagen, CA, USA)) and cDNA was synthesized using ReverTra qPCR RT kit (Toyobo, Osaka, Japan). The relative amount of mRNA expression was analyzed by quantitative PCR (qPCR) with THUNDERBIRDTM SYBR qPCR Mix (Toyobo, Osaka, Japan) according the protocol provide by the manufacturer. The results were normalized by the 18S or Gapdh expression levels. See Appendix A for primer sequences.

### 2.6. Semi-Qunatitative RT-PCR

cDNA was diluted to 30 ng/μL and PCR reactions were optimized to 95 °C for 5 min, amplification cycles (Npy, Npy 1r-40, Gapdh-36) at 95 °C for 30 s, the appropriate annealing temperature (Npy-53 °C, Npy 1r-53 °C, Gapdh-60 °C) for 30 s, 72 °C for 40 s, and a final extension of 7 min at 72 °C. Amplified products were resolved on 3% agarose gels and visualized by SYBR Gold staining.

### 2.7. Histological Analyzes

Adipose tissue and liver were fixed in 4% paraformaldehyde (PFA), embedded in paraffin, and stained with hematoxylin and eosin (H&E) staining. After staining, Images were acquired on a BZ-X700 (Keyence, Osaka, Japan). 4 animals per group were analyzed. Adipocyte size and brown adipocyte number were measured by an application of BZ-X700 and Image J (Analyzed area: 0.2 mm^2^).

### 2.8. Cell Culture

The 3T3-L1 cells were cultured in Dulbecco’s modified Eagle’s medium (DMEM) containing with 10% calf serum. Differentiation was induced 2 days after confluence (Day 0) by adding an induction cocktail containing 100 nM insulin, 1 μM dexamethasone, and 0.5 mM 1-methyl-3-isobutyl-xanthine in DMEM/F12 medium containing 10% fetal bovine serum (FBS) for 3 days. On Day 3, differentiation medium was replaced with Dulbecco’s minimal essential media (DMEM)/F12 medium containing 10% FBS and 100 nM insulin for additional 4 days, and then media were changed every 2 days. The RAW264.7 murine macrophage cells were cultured in DMEM containing 10% fetal bovine serum. All media contained 100 U/mL penicillin, and 100 μg/mL streptomycin. All cells were incubated at 37 °C in a humidified 5% CO_2_. When the cells reached confluence, they were harvested with a cell scraper and diluted with fresh complete medium.

### 2.9. Primary Peritoneal Macrophage Preparation

5 mL of ice cold PBS with 3% fetal calf serum (FCS) was administrated into the peritoneal cavity using a 27-G needle. A 25-G needle attached to a 5 mL syringe was inserted in the peritoneum and collected the fluid. The fluid was then centrifuged at 500× *g* 4 °C for 10 min. Discard supernatant and resuspend cell pellet in DMEM/F12 medium with 10% FCS, 0.2 units/mL penicillin, 100 µg/mL streptomycin, 2 mM L-glutamine, and 25 mM glucose. Red blood cells were lysed before cell culture.

### 2.10. Co-Culture of 3T3-L1 Adipocytes and Macrophages

3T3-L1 cells (2 × 10^5^) were seeded in to 12-well plate, and differentiated for 7 days. Indirect (transwell) coculture was performed by incubating peritoneal macrophages (3 × 10^5^ cells) from NPY^+/+^ and NPY^−/−^ mice-fed HFD in 8-um-pore-size cell culture inserts (BD Bioscience) and placing them in 12-well plates containing differentiated 3T3-L1 adipocytes. Co-cultures were incubated for 2 days and isolated RNA for qPCR.

### 2.11. Statistical Analysis

All data are presented as means ± SEM and were analyzed by unpaired two-tailed Student’s *t* test and two-way analysis of variance (ANOVA) with Bonferroni’s multiple-comparisons test. Statistical analysis was performed with GraphPad Prism 5 and *p* < 0.05 was considered significant.

## 3. Results

### 3.1. Deletion of NPY in Mouse Protects against High-Fat Diet (HFD)-Induced Obesity

Male NPY^+/+^ wild type (WT) mice fed an HFD has grown significantly more compared to the mice on the standard CRF1. Moreover, NPY^−/−^ knockout (KO) mice on an HFD had gained significantly less weight compared to WT mice after 24 weeks (Figure 1A). The lower weight gain was manifested in the decreased accumulation of both visceral (vAT) and subcutaneous fat (sAT, Figure 1D), but these decreased body weight gain and WAT mass were not due to the differences in food intake (Figure 1B). Histological analysis of epididymal and inguinal white adipose tissue (eWAT and iWAT, respectively) from HFD mice showed that white adipocyte size was small in KO mice compared to WT mice (Figure 1E,F). Furthermore, the KO mice had smaller lipid droplets in brown adipocytes than the WT mice after feeding an HFD (Figure 1E,G). Interestingly, NPY deficiency mainly inhibited the HFD-induced decrease in the number of brown adipocytes (Figure 1G). Consistent with the decreased adiposity, serum free fatty acids (FFA) levels were significantly lower in KO mice than WT mice (Figure 1C). However, body weight, fat amounts, and FFA levels in mice on the CRF1 diet were similar between the WT and KO mice. These results indicate that NPY deficiency diminished the HFD-induced increase in body weight and fat mass.

### 3.2. Loss of NPY Results in an Alteration of the Energy Source

Next, we assessed the metabolic effects in mice on an HFD. Total oxygen consumption (VO_2_) was not significantly altered between the KO and WT mice (Figure 2A). We also measured the respiratory quotient (RQ = volume of carbon dioxide produced (VCO_2_)/volume of oxygen consumed (VO_2_). The RQ in dark cycle was lower than the light cycle in WT mice, while, the RQ was low trend in light cycle of KO mice compared to that of WT mice and was similar between light and dark cycle in KO mice (Figure 2B). These results suggest that high lipid to carbohydrate oxidation in the light cycle could contribute to reduced fat accumulation in KO mice fed an HFD.

### 3.3. Deletion of NPY Partly Activates Thermogenesis in BAT

The low RQ value in KO mice compared to the WT mice suggests a high utilization of fat as an energy source in KO mice. To confirm that NPY deficiency regulates adipose tissue metabolism during an HFD, we measured the mRNA expression of de novo lipogenesis/lipolysis-related genes in WAT. However, the expression of these genes was not altered by NPY deficiency in HFD-fed mice (Appendix A). Next, we examined whether NPY deficiency affected thermogenesis. The expression of uncoupling protein 1 (UCP1), the main regulator of thermogenesis, in eWAT, iWAT, and BAT, was high in HFD-fed WT mice compared to CRF1-fed WT mice (Figure 2C and Appendix A), while it was not altered between KO and WT mice on an HFD The mRNA expression of other thermogenic genes, including Cox8b, Cidea, and Prdm16, was slightly increased in BAT from KO mice than in BAT from WT mice under an HFD (Figure 2D) but did not show any significant differences in iWAT from the KO and WT mice under HFD (Appendix A). These results indicate that UCP1-independent thermogenesis in BAT could contribute to lower fat accumulation in KO mice on an HFD.

### 3.4. NPY Ablation Improves Glucose Tolerance and Attenuates Adipose Tissue Inflammation

We performed an intraperitoneal glucose tolerance test in mice fed CRF1 and HFD for 16 weeks. The HFD mice presented high blood glucose concentrations compared to CRF1 mice for 120 min. In response to a glucose challenge, KO mice cleared glucose from the blood-stream significantly better than WT mice during both CRF1 and HFD-fed conditions (Figure 3A). The initial increase in blood glucose was similar up to 30 min, after which the levels in the KO mice remained lower than in WT mice. The blood insulin level was elevated by HFD, but not altered by NPY deficiency (Figure 3B).

Adipose tissue inflammation is associated with the development of obesity. Excessive adipose tissue expansion leads to adipocyte death, which in turn recruits pro-inflammatory macrophages to the adipose tissues. Infiltration of macrophages contributes to adipose tissue dysfunction through the induction of inflammation during HFD-induced obesity [17]. The monocyte chemotactic protein 1 (MCP-1), a key chemokine involved in the recruitment of macrophages and CD68, a macrophage marker, was significantly elevated in eWAT from HFD-fed WT mice compared to the CRF1-fed WT mice. However, these increase were not observed in KO mice. Adipokines and chemokines secretion in adipose tissue can regulate metabolic homeostasis, while their abnormal secretion causes metabolic dysfunction during obesity [18]. Our results revealed that NPY deficiency reduced the expression of pro-inflammatory mediators, including Tnf-α, Il6, and Ccl2, in eWAT from fed n HFD (Figure 3C). We moved on to study molecular consequences of NPY in a cell-based system using mouse Raw264.7 macrophages stimulated with LPS and INFγ. NPY treatment upregulated the mRNA expression of Tnf-α, Il6, and Ccl2 (Figure 3D).

### 3.5. NPY Regulates Adipocyte-Macrophage Crosstalk

Npy1r (NPY Y1 receptor)-mRNA was expressed in both adipocytes and macrophages. In contrast, Npy-mRNA was expressed only in macrophages (Figure 4A). Moreover, Npy-mRNA expression in macrophages was significantly higher in HFD-fed mice than in the CRF1fed mice (Figure 4B). Notably, NPY administration increased the mRNA expression of genes involved in fatty acid transport, such as Cd36, Fatp1, and Fatp4, in FFA-treated 3T3-L1 adipocytes (Figure 4C). To investigate whether NPY secretion from macrophages can regulate adipocyte metabolism, we co-cultured 3T3L1 adipocytes with peritoneal macrophages isolated from WT and KO mice fed an HFD. Adipocytes co-cultured with KO macrophages showed decreased expression of adipogenic/lipogenic genes Pparγ2, Cd36, and Plin1 compared to adipocytes co-cultured with WT macrophages, suggesting that NPY secretion from macrophages influence adipocyte metabolism (Figure 4D).

### 3.6. Inhibition of NPY Ameliorates High Fat Diet-Induced Steatosis

To examine whether NPY deficiency ameliorates hepatic steatosis, we performed H&E staining (Figure 5A, left panel), oil red O staining (Appendix A), and measured the triglyceride (Tg) content (Figure 5A, right panel) in the liver. Deletion of NPY caused a significant decrease in the accumulation of neutral lipids and Tg (Figure 5A and Appendix A). Furthermore, NPY mRNA expression in macrophages positively correlated with the liver weight/body weight ratio (Figure 5B). These results suggest that NPY deficiency improves hepatic lipid metabolism. Next, the mechanisms underlying the attenuation of hepatic steatosis by NPY deficiency were investigated. As shown in Figure 5C, NPY deficiency decreased mRNA expression of hepatic Cd36 and Lpl, which are involved in fatty acid uptake, in HFD-fed mice, while the mRNA levels of other fatty acid uptake genes, including Fapt2 and Fapt5, were not altered (Figure 5C). Furthermore, the expression of genes associated with Tg synthesis, including Dgat2 and Cyp4a14, was downregulated by NPY deficiency in HFD-fed mice (Figure 5D). In contrast, the mRNA levels of genes involved in de novo lipogenesis/very-low-density lipoprotein (VLDL) secretion/fatty acid oxidation, including Acaca, Fasn, Apob, and Acox1 were not significantly altered in KO mice compared to WT mice fed an HFD (Figure 5D and Appendix A). Together, these results suggest that NPY deficiency improved HFD-induced hepatic steatosis, possibly through the downregulation of fatty acid transport and Tg synthesis.

## 4. Discussion

A systemic imbalance between energy storage and dissipation can lead to an increase in adipocyte cell size (hypertrophy) and number (hyperplasia), which are distinct characteristics of obesity [19]. Obesity-mediated insulin resistance causes an inability to store fat in adipose tissue, resulting in an increased fat accumulation in other organs, such as the liver, heart, muscle, and pancreas. In this study, we demonstrated that deletion of NPY attenuated HFD-induced weight gain and fat accumulation. The hyperplasia and hypertrophy of white adipocytes caused by HFD were remarkably diminished by the deletion of NPY in both eWAT and iWAT. Furthermore, NPY deficiency ameliorated HFD-induced fatty liver by regulating genes involved in fatty acid uptake and Tg synthesis, including Cd36, Lpl, Dgat2, and Cyp4a14. Overall, our results highlight the beneficial effects of NPY antagonism on the regulation of adiposity.

Another key feature of obesity is the chronic inflammatory state of the adipose tissue. Inflammation occurs in the expanding adipose tissue and is associated with the infiltration of immune cells, such as macrophages and lymphocytes, is accompanied by the overproduction of cytokines, such as TNF-α and IL6 [20], while the neutralization of TNF-α in obese mice improves insulin sensitivity and glucose tolerance [21]. In this study, NPY expression in macrophages was dramatically increased by an HFD. In addition, HFD-induced mRNA expression of the macrophage marker, Cd68 and inflammatory genes, including Tnf-α, Il6, Ccl2, and Ccl3 in eWAT, were significantly attenuated by NPY deficiency. Furthermore, experiments using Raw264.7 macrophage demonstrated that NPY augmented the LPS/INFγ-mediated inflammatory signaling pathway. The results of the present study are consistent with other reports that NPY is related to TNFα-mediated inflammation [22,23]. However, the molecular pathways for the HFD-induced NPY and NPY-TNF-α interaction remain unclear and should be further investigated. In co-culture with macrophages and adipocytes, we observed that mRNA expression of genes involved in lipogenesis, such as Pgarγ2, Cd36, and Plin1, was lower in co-culture with fully differentiated 3T3-L1 adipocytes and macrophages from KO mice than that in WT mice. Our results suggest that macrophage-expressed NPY may crosstalk with adipocytes, thereby increasing lipogenesis. We used peritoneal macrophages derived from mice instead of adipose tissue macrophages because the number of adipose tissue macrophages is too less for co-culture experiments with adipocytes. However, the effect of macrophage-expressed NPY in the regulation of adipocyte metabolism was elucidated, although the source of NPY was not adipose tissue macrophages. Thus, the effect of NPY from adipose tissue macrophages warrants further investigation.

CD36 is a multifunctional signaling molecule with several ligands, including long-chain FFA, HDL, LDL, and VLDL [24], and plays a pivotal role not only in adipose inflammation but also in liver steatosis [25,26]. Notably, we observed that NPY deficiency inhibited blood FFA and CD36 expression in both WAT and liver. These findings suggest that fatty acid metabolism regulation by NPY deficiency might have inhibited HFD-induced obesity and NAFLD pathogenesis. De novo lipogenesis is an intracellular mechanism that triggers fatty acid synthesis from glucose. De novo lipogenesis is downregulated in WAT from obese mice and is associated with enhanced glucose tolerance and insulin sensitivity [27]. However, in our study, mRNA expression of genes involved in de novo lipogenesis, such as ChREBP-α, Acaca, and Fasn, was not altered in eWAT from KO mice compared to that in WT mice.

Activation of thermogenesis in brown or brite (also named beige) adipocytes is inversely correlated with body weight and adiposity, making brown and brite adipocytes a potential target for obesity therapeutics [28,29]. Although BAT was once considered necessary only in early neonates, several studies showed that BAT was detectable in adult cervical–supraclavicular regions and thoracic and abdominal para-spinal sites and plays an important role in energy balance [30,31]. The non-shivering thermogenic regulator, UCP1, is a mitochondrial protein specific to BAT and that uncouples cellular respiration and mitochondrial ATP synthesis to dissipate energy in the form of heat. We recently found that the thermogenic program in male mice fed a control diet was not activated by NPY deficiency, but this was not investigated in mice fed an HFD [32]. In the current study, we found that mRNA expression of Ucp1 was not increased by NPY deficiency, although mRNA expression of other thermogenic genes was partly upregulated in BAT from HFD mice. Recent evidence supports the existence of UCP1-independent mechanisms to regulate thermogenesis [33,34], suggesting that deletion of NPY may affect noncanonical UCP1 independent mechanisms such as sarco-endoplasmic reticulum ATPase. The underlying mechanism by which NPY deficiency attenuates UCP1-independent thermogenesis remains unclear and warrants further investigation. Although energy expenditure in both dark and light cycles was not significantly altered by NPY deficiency, we noted a low RQ in KO-fed HFD mice compared to WT-fed HFD mice, especially in the light cycle. Perhaps a high lipid to carbohydrate oxidation for BAT thermogenesis in the light cycle could contribute to reduced fat accumulation in KO mice fed an HFD.

NPY has been considered to regulate adipose tissue metabolism through the neuroendocrine route via the sympathetic nervous system (SNS) [35,36], while it seems to be regulated by paracrine pathways through immune cells [37]. NPY expression in the hypothalamic arcuate nucleus is increased in obesity, which inhibits the SNS through the neuroendocrine route, where it co-exist with norepinephrine. This led to the inhibition of lipolysis and enhanced adipogenic/lipogenic action in WAT. Furthermore, it also decreases BAT activity, thereby reducing thermogenesis [35]. NPY deficiency may promote thermogenic action in BAT by activating the SNS during HFD. Interestingly, our data showed that NPY expression was markedly increased by HFD in macrophages. This result was consistent with the finding of Singer et al., who demonstrated that adipose tissue macrophages influence the adipose tissue environment [37]. In addition, NPY increased the expression of pro-inflammatory genes in macrophages, and its deletion suppressed the expression of lipogenic genes, suggesting that an increase in NPY in macrophages caused by HFD may induce dysfunction of WAT via paracrine pathways.

## 5. Conclusions

In summary, this study provides a link between obesity, NAFLD, and NPY. The expression NPY-mRNA in macrophages is increased by HFD, which promotes metabolic inflammation to accelerate the development of obesity and NAFLD. Mice with NPY deletion were protected against HFD-induced weight gain and fat accumulation, and also showed improved glucose tolerance, reduced inflammation in WAT, and enhanced metabolic capacity of BAT (Figure 6). Thus, NPY antagonism in macrophages may represent a potential therapeutic target for the treatment of obesity and fatty liver disease.

## Figures and Tables

**Figure 1 biomedicines-09-01739-f001:**
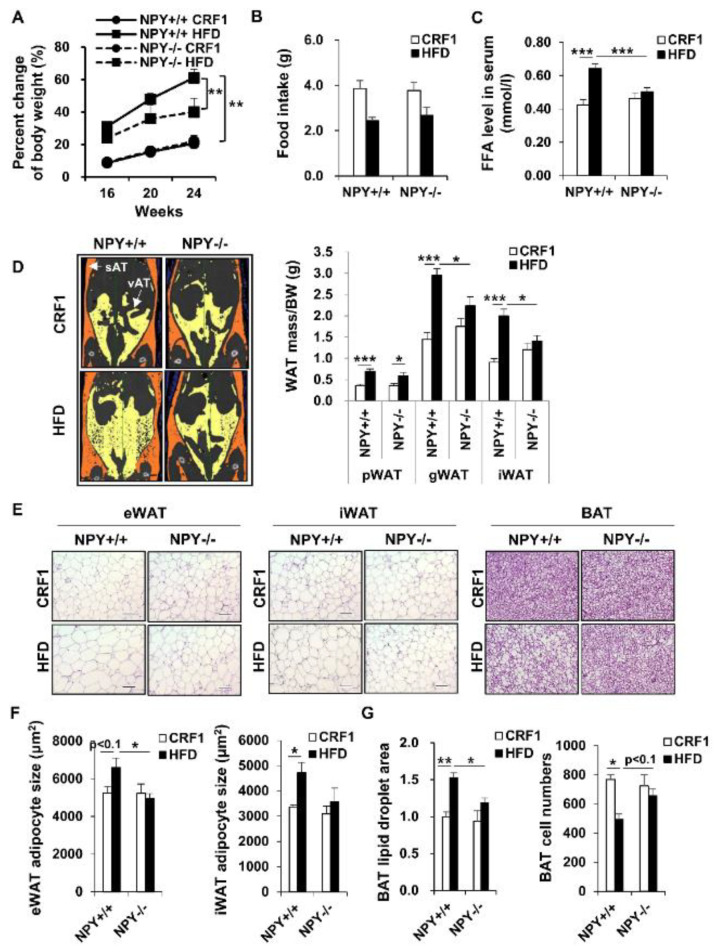
Neuropeptide Y (NPY)^−/−^ mice are resistant to high fat diet (HFD). (**A**) Body weight change during 8 weeks HFD feeding in WT and KO mice. (**B**) Food intake, (**C**) FFA level, (**D**) Representative images of subcutaneous and visceral adipose tissue were determined by 3D-Micro CT (left); perirenal (p), epididymal (e) inguinal adipose tissue (iWAT) mass upon sacrifice (right), (*n* = 6–12 per group), (**E**) Representative H&E (hematoxylin and eosin) staining of eWAT, iWAT, and BAT. Scale bar, 100 μm, (**F**) The mean sizes of adipocyte from eWAT and iWAT (**G**) BAT cell number (*n* = 4 per group), All data are presented as the mean ± SEM, Student’s two-tailed *t* test, * *p* < 0.05, ** *p* < 0.01, and *** *p* < 0.001.

**Figure 2 biomedicines-09-01739-f002:**
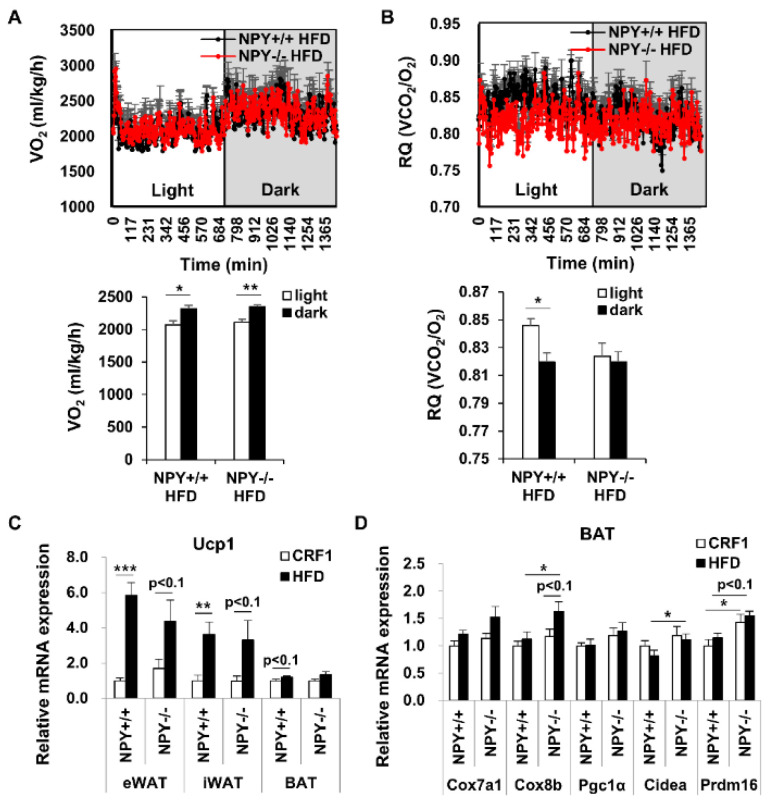
NPY deficiency enhances brown adipose tissue (BAT) thermogenesis. (**A**) Oxygen consumption (botom: average oxygen consumption), (**B**) Respiratory quotient consumption (botom: average respiratory quotient) for 24 h in WT and KO mice-fed HFD (*n* = 4 per group), (**C**) mRNA expression level of UCP1 in eWAT, iWAT and BAT, (**D**) mRNA expression level of thermogenesis-related genes in BAT from WT and KO mice (*n* = 5–6 per group). All data are presented as the mean ± SEM. Statistical significance was determined by two-way analysis of variance (ANOVA) with Bonferroni’s multiple-comparisons test or Student’s two-tailed *t* test, * *p* < 0.05, ** *p* < 0.01, and *** *p* < 0.001.

**Figure 3 biomedicines-09-01739-f003:**
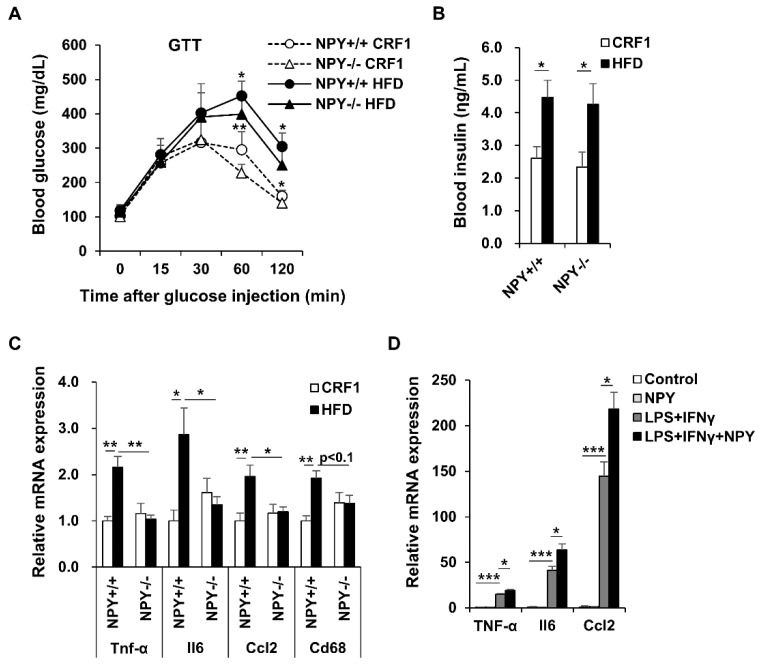
NPY deficiency inhibits inflammation and glucose intolerance. (**A**) (GTT), (**B**) Insulin, (**C**) mRNA expression level of pro-inflammatory gene expression in eWAT from WT and KO mice (*n* = 6–12 per group), (**D**) Effect of NPY on Lipopolysaccharide (LPS) and Interferon gamma (IFNγ)-induced pro-inflammatory gene expression in Raw264.7 cells (*n* = 4 per group). Raw264.7 cells were incubated with 100 nM NPY in the presence or absence of 10 ng/mL LPS and 10 ng/mL INFγ for 24 h. All data are presented as the mean ± SEM, Student’s two-tailed *t* test, * *p* < 0.05, ** *p* < 0.01, and *** *p* < 0.001.

**Figure 4 biomedicines-09-01739-f004:**
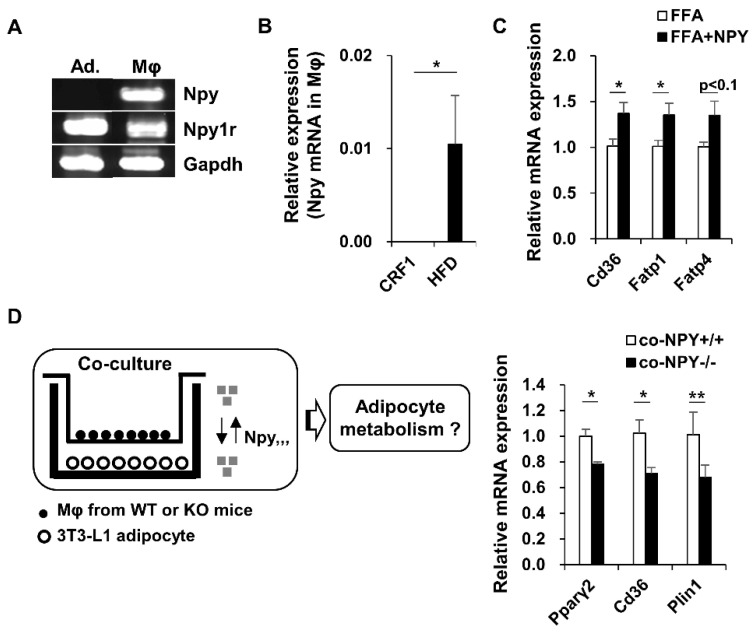
NPY is increased in macrophage in HFD-fed mice and involved in adipogenic and lipogenic genes expression in adipocyte. (**A**) mRNA expression of NPY and Y1R was measured by PCR in 3T3-L1 adipocytes (Ad.) and peritoneal macrophages (Mφ) from mice, (**B**) mRNA expression of NPY in peritoneal macrophage from mice-fed CRF1 and HFD, (**C**) Differentiated 3T3-L1 adipocytes were treated with NPY and FFA (500 μM Palmitate), then analyzed mRNA expression of fatty acid transporters. (**D**) Differentiated 3T3-L1 adipocytes were co-cultured with peritoneal macrophages from WT and KO mice, then analyzed mRNA expression of adipogenic and lipogenic genes. All data are presented as the mean ± SEM (*n* = 4–6 per group), Student’s two-tailed *t* test, * *p* < 0.05, ** *p* < 0.01.

**Figure 5 biomedicines-09-01739-f005:**
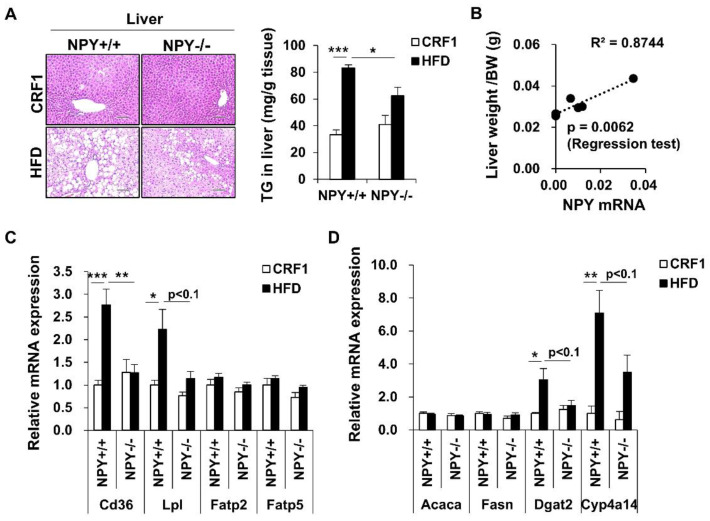
Deletion of NPY improves HFD-induced liver steatosis. (**A**) Representative H&E (hematoxylin and eosin) staining of liver. Scale bar, 100 μm (left panel), liver triglyceride level in WT and KO mice (right panel), (*n* = 4 per group), (**B**) Correlation of NPY mRNA levels in peritoneal macrophage with liver mass/BW ratio from mice fed-HFD, (**C**,**D**) mRNA expression level genes involved in fatty acid uptake and lipogenesis in liver from WT and KO mice. All data are presented as the mean ± SEM (*n* = 5–6 per group). Student’s two-tailed *t* test, * *p* < 0.05, ** *p* < 0.01, and *** *p* < 0.001.

**Figure 6 biomedicines-09-01739-f006:**
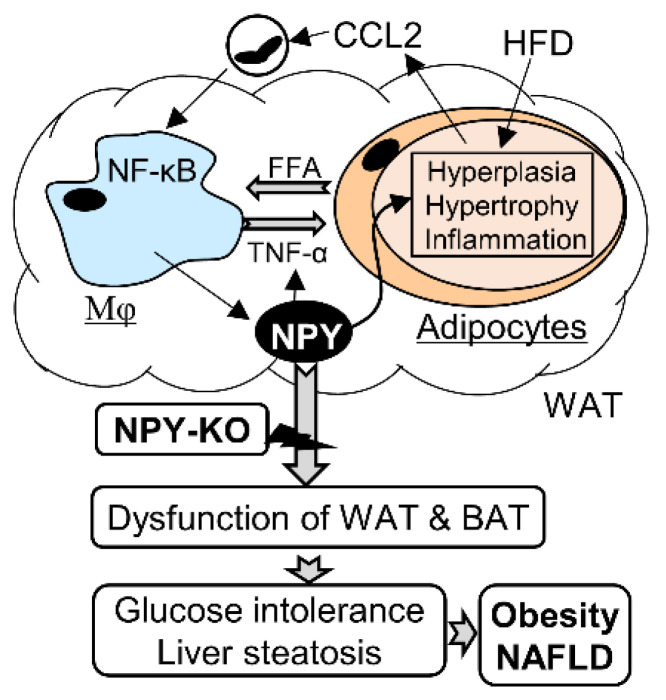
NPY antagonism ameliorates HFD-induced obesity and NAFLD pathogenesis. The expression of macrophage (Mφ) NPY is increased by HFD that promotes adiposity and TNFα-mediated inflammation through CCL2-meidated macrophage infiltration to WAT, leading to development of obesity and NAFLD. Mice with deletion of NPY are protected against HFD-induced dysfunction of WAT and BAT through control of inflammation and fat metabolism thereby ameliorating obesity and NAFLD.

## Data Availability

The data presented in this study are available in insert article or supplementary material here.

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
