# Peer review of "The Role of Neuropeptide Y in Adipocyte-Macrophage Crosstalk during High Fat Diet-Induced Adipose Inflammation and Liver Steatosis"

_biomedicines, 2021, doi:10.3390/biomedicines9111739_

Round 1

Reviewer 1 Report

The manuscript by Park and colleagues describes how the deletion of neuropeptide Y (NPY) attenuated high-fat diet (HFD)-induced weight gain and fat accumulation. Authors observed that HFD-fed mice upregulates NPY-mRNA expression in macrophages. Interestingly they found reduced adipose tissue inflammation and fat accumulation in the liver of HFD-fed NPY-/- knockout mice, suggesting NPY antagonism in macrophages as a potential therapeutic target for obesity and fatty liver disease. This is a potentially interesting manuscript, however extra evidences and minor corrections must be provided to make the paper suitable for Biomedicines audience.

Major Comments:

  1. There is a high variability (SEM) in KO mice food intake -CRF1 and HFD diets- (Figure 1B). Include more animals in the analysis.
  2. In addition to BAT cell “number” (Figure 1F), quantify lipid droplet size in BAT analysis (Figure 1E).
  3. Expression levels of UCP1 in iWAT and BAT shown in Figure 2C non-consistent with Figure S2A. Include significance in S2A.
  4. Include control NPY treatment alone in Figure 3D graph.
  5. Include correlation of NPY with liver mass/BW ratio from mice fed-CRF-1 in Figure 5D graph.

Minor issues:

Include in Material & Methods;

  • LPS -define abbreviation-, INFγ, and NPY (stocks solution concentration and source)
  • Co-culture setup
  • Statistical analysis; test used in Figure 5B
  • Lines 85 and 93; Define abbreviations; Glucose tolerance test (GTT). Triglyceride (TG)
  • Line 111; Describe briefly size measurements analysis, e.g. cell areas measurements on image J, number of cells analyzed

Line 141; 12 weeks?  replace by -> 24 weeks

Lines 158 and 159; Include abbreviations; subcutaneous (sAT), visceral (vAT) adipose tissue. Peripheral (pWAT), epididymal (eWAT), inguinal (iWAT)

Lines 160 and 161; gWAT ? -> eWAT, inWAT -> iWAT

Line 163; Include **p  < 0.01 (Figure 1A)

Figure 1B; Food consumption (FC, g) -> Food intake (g)

Figure 1F; Include t-test significance for iWAT in WT vs. KO HFD-fed mice  (not significant; n.s. ?)

Figure 2A, B; VO2 and RQ levels in bottom graphs, are the average of single measurements (Light vs. Dark) from upper graphs?, Specify in text

Figure 3A; Review graph labels for KO CRF1-fed mice

Figure 3D; Replace “Con” label -> Control

Replace p<0.1 graphs labels by ** (Figure 1F, 2C, 3C, 4C, 5C, and S2B)

Include subscripts: CO2 -> CO2, VCO2 -> VCO2, VO2 -> VO2

Correct nano prefix symbol, ηM -> nM, ηg -> ng

Author Response

Dear Reviewer,

We thank the reviewer for the helpful suggestions. We have followed these suggestions and have extensively revised the manuscript, and we hope that the revised manuscript is now suitable for publication in Biomedicines. Please find the attached PDF file.

Sincerely 

Seongjoon Park

Reviewer 2 Report

Reviewer comments for biomedicines-1459090

In this manuscript, the authors report their findings regarding a novel intracellular mechanism of NPY in the regulation of adipocyte-macrophage crosstalk. These results are interesting. However, similar papers have already been published previously. Moreover, many major and minor issues need to figure out. The details are as follows:

  1. These findings is not novel. It is already published that NPY deficiency attenuates metabolic disorder. Adipose tissue macrophages produced NPY for inhibiting obesity and inflammation are published as well. As follows:

Reference 1: Patel HR, Qi Y, Hawkins EJ, Hileman SM, Elmquist JK, Imai Y, Ahima RS. Neuropeptide Y deficiency attenuates responses to fasting and high-fat diet in obesity-prone mice. Diabetes. 2006 Nov;55(11):3091-8. doi: 10.2337/db05-0624. PMID: 17065347.

Reference 2: Singer, Kanakadurga et al. “Neuropeptide Y is produced by adipose tissue macrophages and regulates obesity-induced inflammation.” PloS one vol. 8,3 (2013): e57929. doi:10.1371/journal.pone.0057929

  1. The authors mentioned that NPY enhances CD36-mediated adipocyte fatty acid uptake and TNF-α-mediated macrophage inflammation in the abstract. However, data is not enough to support this conclusion. Namely, CD36 and TNF-α gene silence should be applied; otherwise, we can’t say mediated by CD36 and TNF-α.
  2. In figure 4, only gene expressions of fatty acid transporters were shown. The results of protein level also need to be provided. The ability of free fatty acid uptake should be performed in vitro.
  3. In figure 5, except for hepatic HE staining, oil red O staining or Nile red staining in the liver also need to show.
  4. The procedures of Quantitative Real-time PCR were described in the Material and Methods portion, but the semi-quantitative PCR result was shown in Figure 4A. It’s unmatching, and please correct it.
  5. In Figure 5A, the Unit of TG in liver is wrong, mg/dl should be changed into mg/mg tissue or mg/mg protein.
  6. In Figure 3, the symbol of group NPY-/-CRF1 and group NPY+/+CRF1 is the same. It will make readers confused, and Please revise it.

Author Response

Dear Reviewer,

We thank the reviewer for the helpful suggestions. We have followed these suggestions and have extensively revised the manuscript, and we hope that the revised manuscript is now suitable for publication in Biomedicines.

Round 2

Reviewer 1 Report

Overall, the revised version of the manuscript has clarified most of the issues raised.

Minor issue

Figure S2A: Review y-axis labeling in graphs for Ucp1 quantification and include the western blot method and antibodies used in Material and Methods section.

Author Response

Dear Reviewer,

We thank the reviewer for the helpful suggestions. We have followed these suggestions and have extensively revised the manuscript, and we hope that the revised manuscript is now suitable for publication in Biomedicines.

Manuscript ID: biomedicines-1459090

MS TITLE: The Role of Neuropeptide Y in Adipocyte-Macrophage Crosstalk During High Fat Diet-Induced Adipose Inflammation and Liver Steatosis

Dear Reviewer,

We thank the reviewer for the helpful suggestions. We have followed these suggestions and have extensively revised the manuscript, and we hope that the revised manuscript is now suitable for publication in Biomedicines.

Comments

Overall, the revised version of the manuscript has clarified most of the issues raised.

Minor issue: Figure S2A: Review y-axis labeling in graphs for Ucp1 quantification and include the western blot method and antibodies used in Material and Methods section.

Response: We corrected them in Figure S2A and added western blot method in supplementary information.

Reviewer 2 Report

We fully understand authors did some novel of the study. However, the authors didn't give a convincible answer regarding some previous comments including comments 2, 3, 4. Therefore, I strongly suggest completing the extra experiments for supporting the conclusion.

please refer to previous comments:

Comments 2: The authors mentioned that NPY enhances CD36-mediated adipocyte fatty
acid uptake and TNF-α-mediated macrophage inflammation in the abstract. However, data is not enough to support this conclusion. Namely, CD36 and TNF-α gene silence should be applied; otherwise, we can’t say mediated by CD36 and TNF-α.

Comments 3: In figure 4, only gene expressions of fatty acid transporters were shown. The results of protein level also need to be provided. The ability of free fatty acid uptake should be performed in vitro.

Comments 4: In figure 5, except for hepatic HE staining, oil red O staining or Nile red staining in the liver also need to show.

Author Response

Dear Reviewer,

We thank the reviewer for the helpful suggestions. We have followed these suggestions and have extensively revised the manuscript, and we hope that the revised manuscript is now suitable for publication in Biomedicines.

Manuscript ID: biomedicines-1459090

MS TITLE: The Role of Neuropeptide Y in Adipocyte-Macrophage Crosstalk During High Fat Diet-Induced Adipose Inflammation and Liver Steatosis

Dear Reviewer,

We thank the reviewer for the helpful comments.

Comments

We fully understand authors did some novel of the study. However, the authors didn't give a convincible answer regarding some previous comments including comments 2, 3, 4. Therefore, I strongly suggest completing the extra experiments for supporting the conclusion.

Comments 2: The authors mentioned that NPY enhances CD36-mediated adipocyte fatty
acid uptake and TNF-α-mediated macrophage inflammation in the abstract. However, data is not enough to support this conclusion. Namely, CD36 and TNF-α gene silence should be applied; otherwise, we can’t say mediated by CD36 and TNF-α.

Response: We also agree to reviewer’s suggestion. The gene silencing experiment using CD36 or TNF-α-siRNA make strongly support that NPY enhances CD36-mediated adipocyte fatty acid uptake and TNF-α-mediated macrophage inflammation. As we explained at first revision, it’s problem of time. We have tissues and cells to do those experiments, but it is impossible to do in a second revision deadline (7 days). Please understand this situation. Therefore, we revised our abstract. We modified the abstract to avoid an exaggerated interpretation.

Comments 3: In figure 4, only gene expressions of fatty acid transporters were shown. The results of protein level also need to be provided. The ability of free fatty acid uptake should be performed in vitro.

Response: As reviewer’s suggestion, protein levels fatty acid transporters can support the mRNA expression results, but it is also same as above response. we don’t have enough time to figure it out until second revision deadline (7 days). Please understand this situation.

Comments 4: In figure 5, except for hepatic HE staining, oil red O staining or Nile red staining in the liver also need to show.

Response: As reviewer’s suggestion, we added result of oil red O staining in Figure S3 and added methods in supplementary information.

Round 3

Reviewer 2 Report

No more comments, Appreciate your response in detail.